# Transcriptomic Analysis of the Porcine Gut in Response to Heat Stress and Dietary Soluble Fiber from Beet Pulp

**DOI:** 10.3390/genes13081456

**Published:** 2022-08-16

**Authors:** Minju Kim, Eugeni Roura, Yohan Choi, Joeun Kim

**Affiliations:** 1Centre for Nutrition and Food Sciences, Queensland Alliance for Agriculture and Food Innovation, The University of Queensland, Brisbane, QLD 4072, Australia; 2Swine Science Division, National Institute of Animal Science, Rural Development Administration, Cheonan 31000, Korea

**Keywords:** pig, RNA-Seq, heat stress, small intestine, gene expression, soluble fiber, beet pulp

## Abstract

This study aimed to investigate the impact of heat stress (HS) and the effects of dietary soluble fiber from beet pulp (BP) on gene expression (differentially expressed genes, DEGs) of the porcine jejunum. Out of the 82 DEGs, 47 genes were up-regulated, and 35 genes were downregulated between treatments. The gene ontology (GO) enrichment analysis showed that the DEGs were related mainly to the actin cytoskeleton organization and muscle structure development in biological processes, cytoplasm, stress fibers, Z disc, cytoskeleton, and the extracellular regions in cellular composition, and actin binding, calcium ion binding, actin filament binding, and pyridoxal phosphate binding in the molecular function. The KEGG pathway analysis showed that the DEGs were involved in hypertrophic cardiomyopathy, dilated cardiomyopathy, vascular smooth muscle contraction, regulation of actin cytoskeleton, mucin type O-glycan biosynthesis, and African trypanosomiasis. Several of the genes (*HSPB6*, *HSP70*, *TPM1*, *TAGLN*, *CCL4*) in the HS group were involved in cellular oxidative stress, immune responses, and cellular differentiation. In contrast, the DEGs in the dietary BP group were related to intestinal epithelium integrity and immune response to pathogens, including *S100A2*, *GCNT3*, *LYZ*, *SCGB1A1*, *SAA3*, and *ST3GAL1*. These findings might help understand the HS response and the effect of dietary fiber (DF) regarding HS and be a valuable reference for future studies.

## 1. Introduction

Heat stress (HS) is increasingly perceived as a critical issue in the livestock industry due to the increasing environmental temperatures caused by global warming. The annual economic losses due to HS reach almost USD 2.36 billion, as estimated by the U.S livestock industry alone [1]. Pigs are particularly vulnerable to HS as they have less functional sweat glands and a thick subcutaneous adipose layer, which interrupts effective heat loss [2]. In addition, previous studies have shown that HS impacts the porcine gastrointestinal tract by decreasing intestinal integrity and barrier function, and the absorptive and digestive ability, which in turn results in diminishing growth performance and carcass composition [3,4,5,6,7].

Nutritional interventions can be one of the strategies to minimize the impact of HS. The supplementation with micro-nutrients, such as vitamins and trace minerals, and the adjustment of dietary nutrient composition, such as crude protein, fat, and fiber, showed a positive effect on growth performance and productivity in pigs following HS [8,9,10,11,12,13,14,15,16]. In addition, dietary fiber (DF) has been shown to affect the barrier function, the intestinal immune system, and the production of short-chain fatty acids (SCFAs) in animals under HS [17,18,19]. Furthermore, the gut microbiota rapidly fermented the soluble fiber due to its high capacity for hydration-releasing SCFAs, such as acetate, propionate, and butyrate, which play an influential role in maintaining gut homeostasis, and mucus production and secretion [20]. Thus, soluble fiber has the potential to contribute to the maintenance of gut health and integrity following HS [21,22].

Previous studies have demonstrated that HS impacts on the porcine intestine and effects a complex cellular response, including altering the expression of various genes, resulting in the modulation of biological processes and metabolic pathways, including the unfolding of proteins, initiation of translation, and cell proliferation and migration [23,24,25]. In addition, a high DF (6.5%) increased the expression of the glucose transporters 3 and 4, while downregulating the heat shock protein 70, during HS [26]. However, these studies were limited to individual genes and pathways, failing to report a more holistic view on the complex network of biological responses involved in HS and DF in pigs. Transcriptome sequencing facilitates the identification of the entire genome expression, leading to a better understanding of the HS responses and the potential modulatory effects of DF. Therefore, the objective of this study was to perform a transcriptome analysis of the intestinal responses to heat stress in pigs fed dietary soluble fiber, such as sugar beet pulp (BP).

## 2. Materials and Methods

This study was conducted at the facility of the National Institute of Animal Science following the protocol of the Institutional Animal Care and Use Committee at the National Institute of Animal Science (Project No. PJ014796).

### 2.1. Animals and Treatments

A total of 24 barrows (Yorkshire × Landrace × Duroc; average initial BW, 56.71 ± 1.74 kg) were used in the present study and randomly assigned to three treatments consisting of the optimal temperature (NT, *n =* 8), HS (HS, *n =* 8) or HS with dietary supplemental BP (HS + BP, *n =* 8). The NT treatment had an environment that kept the temperature at 23 °C and 70% relative humidity on average for the experimental periods. The animals in the HS treatment were exposed to a 32 °C temperature and 85% relative humidity on average during the feeding trial (Appendix A). BP was supplemented with a 40 g/kg diet for the HS + BP treatment. The experimental period lasted 56d (28d in the grower phase and 28d in the finisher phase). The basal diet was formulated to meet or exceed the nutrient requirements of NRC [27] for growing-finishing pigs (Appendix A). During the overall experimental period, the pigs were allowed ad libitum access to water and the diets.

### 2.2. Sample Collection

At the end of the feeding trial in the experimental period, three pigs per treatment were selected and euthanized to collect the gut samples. The abdomen was immediately opened, and the intestinal tissue samples were removed and flushed with ice-cold phosphate-buffered saline (PBS). Subsequently, about 500 mg of jejunum portion at the middle of the total small intestine’s length were collected and promptly frozen in liquid nitrogen and stored at −80 °C until required for analysis.

### 2.3. RNA Isolation

The total RNA was isolated using Trizol reagent (Invitrogen). The RNA quality was assessed by the Agilent 2100 bioanalyzer (Agilent Technologies, Amstelveen, The Netherlands), and the RNA quantification was performed using the ND-2000 Spectrophotometer (Thermo Inc., Madison, WI, USA).

### 2.4. Library Preparation and Sequencing

The libraries were prepared from total RNA, using the NEBNext Ultra II Directional RNA-Seq Kit (New England Biolabs Ltd., Hitchin, UK). The mRNA isolation was performed using the Poly(A) RNA Selection Kit (LEXOGEN, Inc., Vienna, Austria). The isolated mRNAs were used for the cDNA synthesis and shearing, following the manufacturer’s instructions. The indexing was performed using the Illumina indexes 1-12. The enrichment step was carried out using PCR. Subsequently, the libraries were checked using the TapeStation HS D1000 Screen Tape (Agilent Technologies, Amstelveen, The Netherlands) to evaluate the mean fragment size. The quantification was performed using the library quantification kit using a StepOne Real-Time PCR System (Life Technologies, Grand Isle, NY, USA). High-throughput sequencing was performed as paired-end 100 sequencing using NovaSeq 6000 (Illumina Inc., San Diego, CA, USA).

### 2.5. Quality Analysis and Mapping of Reads

The quality control of the raw sequencing data was performed using FastQC [28]. The adapter and low-quality reads (<Q20) were removed using FASTX_Trimmer [29] and BBMap [30]. Then, the trimmed reads were mapped to the NCBI Genome ***Sus scrofa*** using TopHat [31]. The RC (Read Count) data were processed based on the FPKM+Geometric normalization method, using EdgeR within R [32]. The FPKM (Fragments Per kb per Million reads) values were estimated using Cufflinks [33]. The data mining and graphic visualization were performed using ExDEGA (Excel-based Differentially Expressed Gene analysis) software (E-biogen Inc., Seoul, Korea).

### 2.6. Bioinformatics Analysis

The differential gene expression analysis was performed with ExDEGA v1.6.8 software (e-Biogen, Seoul, Korea) and using a cutoff at the normalized gene expression (log2) of 4 and *p*-value of <0.05. The DEGs were identified based on >2.0-fold change observed in the transcript levels. After filtering the DEGs, the GO annotation analysis was performed using the DAVID bioinformatics program (https://david.ncifcrf.gov accessed on 1 March 2022) for gene identification and annotation. The RNA-Seq data were further applied, using the Kyoto Encyclopedia of Genes and Genomes (KEGG) database (www.genome.jp accessed on 1 March 2022).

## 3. Results

### 3.1. Screening and Clustering of the Gut Affected by HS and Dietary Supplemental BP

The gene expression was compared between the treatment under a normal temperature (NT) vs. the treatment under HS (HS), between the NT and the treatment fed the diet supplemented with BP under HS (HS + BP), and between the HS and the HS + BP. The genes with a ≥2-fold change value were used to generate a scatter plot between the treatments, as shown in Figure 1A,B. The red and green dots indicate the upregulated and downregulated genes, respectively, whereas the black dots mean non-DEGs. The principal component analysis (PCA) using DEGs showed the first and second contribution rates were 47% and 23%, respectively, and the separation between the heat-stressed and normal temperature groups (Figure 1C). The RNA-Seq analysis found a total of 82 DEGs among the three experimental treatments. The HS compared to the NT resulted in the upregulation of 31 genes and the downregulation of 30 genes, whereas the group with the dietary supplemental BP under HS had 42 upregulated and 20 downregulated genes compared to the NT group. The effect of the dietary BP under HS was assessed by comparing the non-beet-pulp HS to the HS + BP groups, resulting in six genes upregulated and six genes downregulated. The part correlation between the HS and HS + BP showed 26 genes and 15 genes in upregulation and downregulation, respectively. The DEGs observed for the pairs of the three experimental groups are compared in the Venn diagram shown in Figure 1D. To obtain a deeper understanding of the gene expression patterns of the porcine gut in the normal temperature group, heat-stressed group, and dietary BP-fed group, a hierarchical cluster analysis of these DEGs was conducted. The results indicated three clusters associated with the three treatments, with three distinct gene expression patterns (Figure 2).

### 3.2. Porcine Gut Transcriptome Response to HS

The response to HS was indicated according to the DEGs identified in the part correlation between the HS vs. NT and the HS + BP vs. NT. The HS resulted in 23 upregulated genes and 11 downregulated genes being significantly (*p* < 0.05) differentially expressed. Among the top upregulated DEGs in the HS and HS + BP were *GSTA1* (glutathione S-transferase α 1), *CKM* (creatine kinase, M-type), *TAGLN* (transgelin), *TPM2* (tropomyosin 2, β), *PCP4* (purkinje cell protein 4), *DES* (desmin), *CNN1* (calponin 1), and *ACTG2* (actin γ 2, smooth muscle), while the top downregulated DEGs included *GZMB* (granzyme B), *PMP22* (peripheral myelin protein 22), *MX1* (MX dynamin-like GTPase 1), *C14H10orf99* (chromosome 14 C10orf99 homolog), and *GNLY* (granulysin) (Table 1).

The GO enrichment analysis showed three different categories: biological process; cellular composition; and molecular function, presented in Figure 3A,B. The enrichment analysis of the significantly DEGs in the biological process showed that among the upregulated DEGs, “actin cytoskeleton organization”, and “muscle structure development” were enriched, whereas “bleb assembly” was enriched amongst the downregulated genes. In the cellular composition, the upregulated DEGs were enriched in “cytoplasm”, “stress fiber”, and “Z disc”. The enrichment of the downregulated genes was associated with “cytoplasm”. The enrichment of the molecular function in the upregulated genes showed that “actin binding”, “calcium ion binding”, and “structural constituent of muscle” were enriched, whereas “pyridoxal phosphate binding” was involved in the enrichment of the downregulated genes. The KEGG pathway enrichment analysis of the HS group DEGs (Figure 3C) showed that the “hypertrophic cardiomyopathy” and the “dilated cardiomyopathy” pathways were enriched.

### 3.3. Porcine Gut Transcriptome Response to BP under HS

The DEGs in response to the BP under heat stress resulted from the HS + BP and HS treatments presented in Table 1. The dietary supplemental BP under HS showed six genes in upregulation and four genes in downregulation; the upregulated genes were *HBB* (hemoglobin, β), *HSPA6* (heat shock protein family A member 6), *S100A2* (S100 calcium binding protein A2), and *LYZ* (lysozyme); in the downregulated genes, *SCGB1A1* (secretoglobin family 1A member 1), *LOC396781* (IgG heavy chain), *SAA3* (serum amyloid A-3 protein), and *ST3GAL1* (ST3 β-galactoside α-2,3-sialyltransferase 1) were included.

The GO enrichment analysis of the significantly DEGs in the cellular composition showed that the “extracellular region” in the BP group was linked to the up- and downregulated genes. In contrast, the enrichment of the upregulated genes in the molecular function was associated with “protein binding”. The KEGG pathway enrichment analysis of the BP group DEGs (Figure 3D) showed that the “mucin type O-glycan biosynthesis” and the “African trypanosomiasis” pathways were enriched.

## 4. Discussion

HS causes adverse impacts on the molecular and cellular pathways, damaging the various metabolic processes with a detrimental impact on animal health and production [34]. HS appears when the heat accumulation in the body exceeds the body’s heat loss [2,22]. The gastrointestinal (GI) tract is one of the main systems affected by HS, causing a decrease in intestinal integrity and barrier function, which in turn may affect the post-absorptive metabolism and tissue accretion [3,5,6,7]. These detrimental effects on the porcine GI tract under heat stress seemed to be associated with differences in the tight junction protein expression, increased the endotoxin concentrations circulating in the body, and increased the cellular oxidative stress [35,36]. Therefore, high-throughput sequencing can shed light on the heat resistance and the genes associated with heat sensitivity in the porcine small intestine. A feasible approach can be provided to find novel insights into the responses to HS in pigs. Therefore, this study aimed to identify the responsive genes and pathways against HS in the porcine jejunum, using transcriptomics analysis. In addition, the most effective functional genes associated with the dietary supplementation of BP and HS were also studied [19]. Three pigs without HS, three under HS, and three fed supplemental BP under HS were selected in this study, from which a total of 30,048 genes were screened out, and 82 DEGs (*p* < 0.05) of those were detected in the RNA-Seq results.

### 4.1. Transcriptome Regulation in Response to HS

The transcriptome analysis in this study showed that 23 genes were found to be significantly regulated in response to HS. Among the DEGs, some were involved in heat shock and stress response. The heat shock proteins (HSPs) are released in response to HS, functioning as molecular chaperones that detect the aggregation of damaged proteins and assist in the refolding of the HS-damaged proteins [37,38,39]. In this study, *HSPB6* and *HSPA6* were upregulated in the heat-stressed groups. The *HSPB6* is a multifunctional protein involved in regulating smooth muscle contraction and demonstrating cardioprotective activity [40]. Under HS, the *HSPB6* is heavily upregulated to maintain the cells’ homeostasis conditions and improve the cell survival functions [41]. The *HSPA6* is the most heat-inducible member of the *HSP70* gene family that plays a role in endoplasmic reticulum chaperone and the sensor of protein misfolding [42,43]. In addition, the expression of the *HSP70* was relative to the *PMP22* gene that was downregulated in the current study. According to a previous study, the proteotoxic stress derived from HS can amplify the inducible *HSP70*, which plays an essential role in reducing the *PMP22* aggregation and preventing the accumulation of the *PMP22* aggregates in the cells [44]. In this process, the *PMP22* gene might be downregulated. *CKM*, associated with intramuscular energy, creatine, and ADP phosphor-exchange, was upregulated in the HS groups. The high levels of *CKM* in the muscle lead to an increase in the rate of muscle metabolism, reducing the pyruvate production and ATP phosphorylation [45,46,47]. In contrast, HS decreased the *CKM* expression in porcine muscle [48]. In addition, the current results showed that *MYLK*, *TPM1*, and *TAGLN* were upregulated, and *CCL4*, *MX1*, and *GNLY* were downregulated by HS. *TPM1* and *TAGLN* are TGFβ-inducible genes known to be involved in actin cytoskeleton organization and cellular differentiation [49,50]. Our results coincide with those of He et al. [51], who found that rat intestinal *TAGLN* expression was upregulated after acute heat shock. *CCL4* is crucial for the immune responses to infection and inflammation [52]. Consistent with our results, *CCL4* was downregulated in the small intestine and in the blood samples of heat-stressed rats and broiler chickens, respectively [53,54]. *GNLY* is a defensin-like cytolytic molecule capable of disrupting bacterial membranes, which had been related to heat shock protein-induced apoptosis [55]. *GSTA1* and *TPM2* were upregulated during cellular oxidative stress, resulting in increased metabolic activity associated with the response to increased oxidative stress [56,57,58]. In this study, the DES gene was upregulated following continuous HS. This result was in accordance with the result that demonstrated that the porcine DES gene involved in the skeletal muscle was significantly upregulated by constant heat stress [59]. Several of the genes involved in cell structure and motility were upregulated in the HS groups, including *MYL9* and *SYNM*. Our current results are consistent with previous studies, as both *MYL9* and *SYNM* were increased with the increase in *PPC1B* due to chronic HS in the porcine small intestine [60,61]. The *PYGM* gene in the porcine small intestine was differentially expressed by HS in this study, and that is an initial enzyme of the glycogenolysis involved in carbohydrate metabolism [62]. A previous study suggested that glycogenolysis was improved, and muscle glycogen mobilization was responsible for the glucose pool available for glycolysis following HS [63].

### 4.2. Transcriptome Regulation in Response to BP under HS

The dietary nutritional strategies may be an effective intervention method to reduce HS. The supplemental micro-nutrients, such as vitamins and trace minerals, and adjusting the dietary nutrient composition, such as crude protein, fat, and fiber, were shown to improve growth performance and productivity in pigs under HS [8,9,10,11,12,13,14,15,16]. BP is rich in soluble digestible fiber, such as pectin, which is rapidly fermented by the gut microbiota due to its high hydration capacity [19,64]. The advantages of dietary supplemental BP are the possibility of alleviating the impact of HS in the porcine small intestine. Our data studied the DEGs in the jejunum of the pigs under HS-fed diets with (HS + BP) or without (HS)-supplemented BP.

Although the dietary supplemental BP did not show the clearly different results in the PCA analysis and did not change the number of DEGs, our results showed some significant changes in the BP group. The small intestine (represented by the jejunum in our study) is the main site for nutrient digestion and absorption. Our results clearly showed that the DEGs associated with BP in the heat-stressed pigs were related to intestinal epithelium integrity and immune response to pathogens, including *S100A2*, *GCNT3*, *LYZ*, *SCGB1A1*, *SAA3*, and *ST3GAL1*. The *S100A2* was upregulated by BP in this study. The *S100A2* functions as a regulator in cell-cycle progression and differentiation [65,66], stimulating the innate immunologic responses [67]. The upregulated *S100A2* gene suggests an increase in the cellular growth or the division processes in the jejunal epithelial. *GCNT3* was upregulated by BP, and the enzyme is known to be involved in mucin biosynthesis, the cell adhesion molecule cadherin-like 26 (*CDH26*), and the immunoregulatory cytokine, IL-13. *GCNT3* has an essential role providing a protective barrier to intestinal pathogens by contributing to the mucus-producing goblet cells and differentiating enterocytes [68,69]. *LYZ* is the Paneth cell-marker involved in crypt differentiation [70]. The upregulation of the *LYZ* gene suggests that the dietary BP triggered the renewal of the intestinal epithelium by increasing the cell proliferation and differentiation. Our results are consistent with the existing literature, showing that the inclusion of high fiber (soluble and/or insoluble) in diets increases the rate of cell proliferation and crypt depth in the intestine [21]. Several of the genes involved in the immune response were downregulated, including *SCGB1A1*, *ST3GAL1*, and *SAA3*. *ST3GAL1* contributes to the synthesis and metabolism of glycolipids, which play a vital role in recognizing the pathogens permeating into intestinal barriers and regulating the CD8+ T lymphocyte homeostasis by modulating O-glycan biosynthesis [71,72]. Our results on *ST3GAL1* may partially contradict the findings by Xia and co-workers, who showed an upregulation of the gene in the ileum of pigs after soluble fiber inulin supplementation [73]. The inconsistency of the results may be related to the large production of the short-chain fatty acids associated with BP fermentation, which in turn could inhibit the *ST3GAL1* expression, as shown by butyrate supplementation in broilers [74]. *SAA3*, which is known as a marker in pathological and adverse physiological conditions, is involved in viral replication, facilitating virus adsorption to the cells [75,76,77]. A previous study reported that the replication of the porcine reproductive and respiratory syndrome virus (PRRSV) uses the *SAA3* protein to adsorb onto the cells [77]. Thus, it is tempting to speculate that the downregulation of *SAA3* by BP supplementation may have a protective effect against PRRSV.

## 5. Conclusions

In conclusion, the results presented using RNA-Seq analysis showed that 23 genes were differentially expressed in the HS response. These genes are mainly involved in biological processes, such as those associated with actin cytoskeleton organization, muscle structure development, cell proliferation, cytoplasm, stress fiber, actin binding, and calcium ion binding. In relation to the effects of the dietary BP, some significantly expressed genes involving in intestinal epithelium integrity and immune system were demonstrated. This transcriptome study provides theoretical evidence for the HS response and strategies to minimize the HS effects. The results of this study improve our understanding of the HS responses and the role of dietary fiber (BP) in pigs.

## Figures and Tables

**Figure 1 genes-13-01456-f001:**
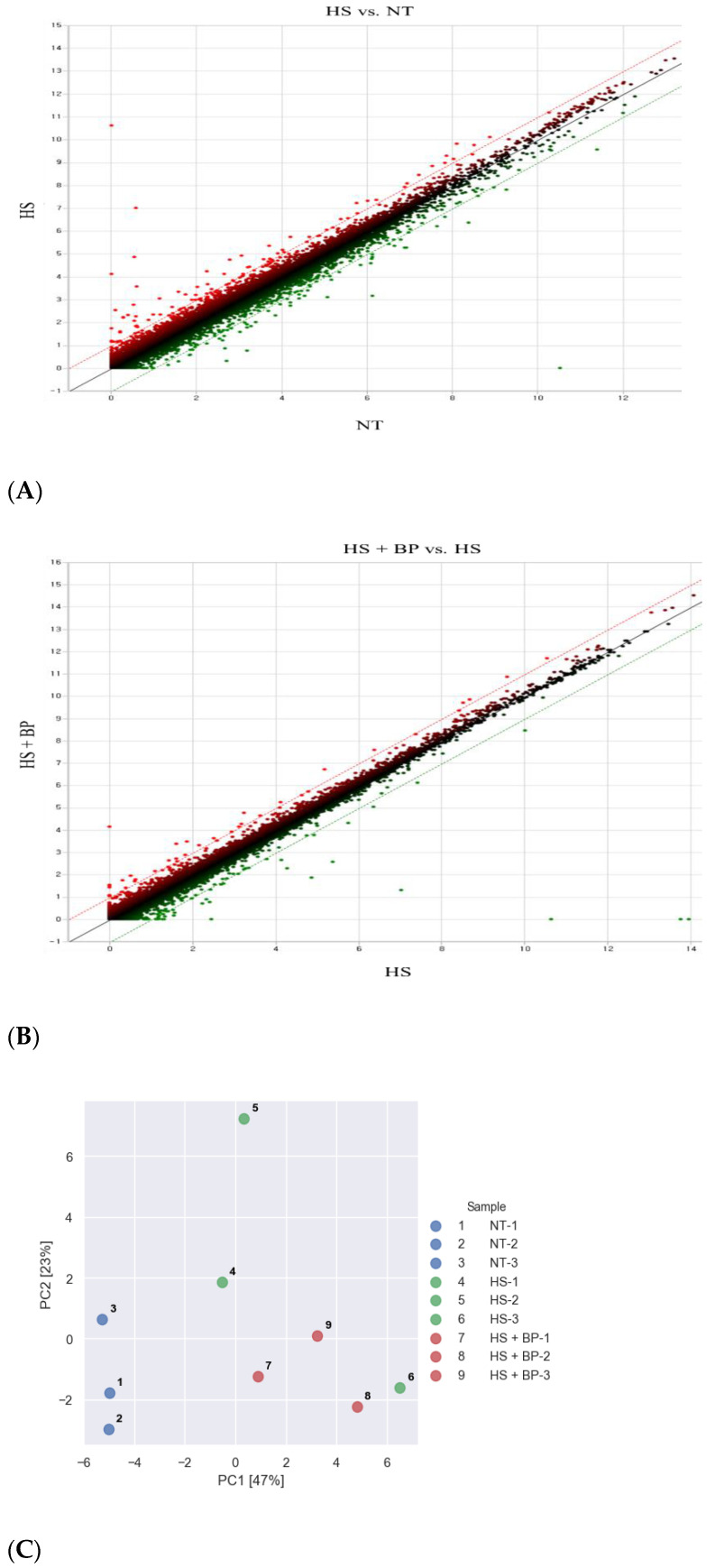
Summary of the DEGs (DEG; *p* < 0.05, ≥2-fold change) in the porcine jejunum under HS and fed dietary BP: (**A**) Scatter plot in HS vs. NT; (**B**) Scatter plot in HS + BP vs. HS; (**C**) PCA analysis based on DEGs of the NT, HS, and HS + BP groups; (**D**) Venn diagrams indicating DEGs; NT, normal temperature; HS, heat stress; HS + BP, dietary beet pulp under heat stress.

**Figure 2 genes-13-01456-f002:**
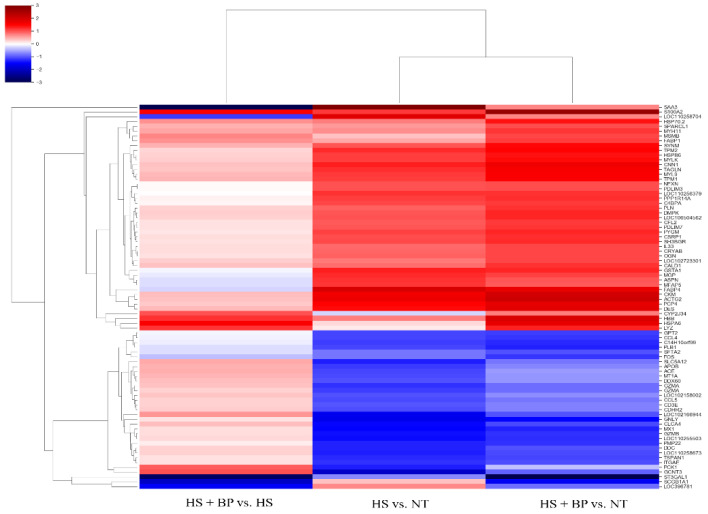
Hierarchical clustering of DEGs in HS vs. NT, HS + BP vs. NT, and HS + BP vs. HS; NT, normal temperature; HS, heat stress; HS + BP, dietary beet pulp under heat stress; Up- and downregulated genes are shown in red and blue color, respectively.

**Figure 3 genes-13-01456-f003:**
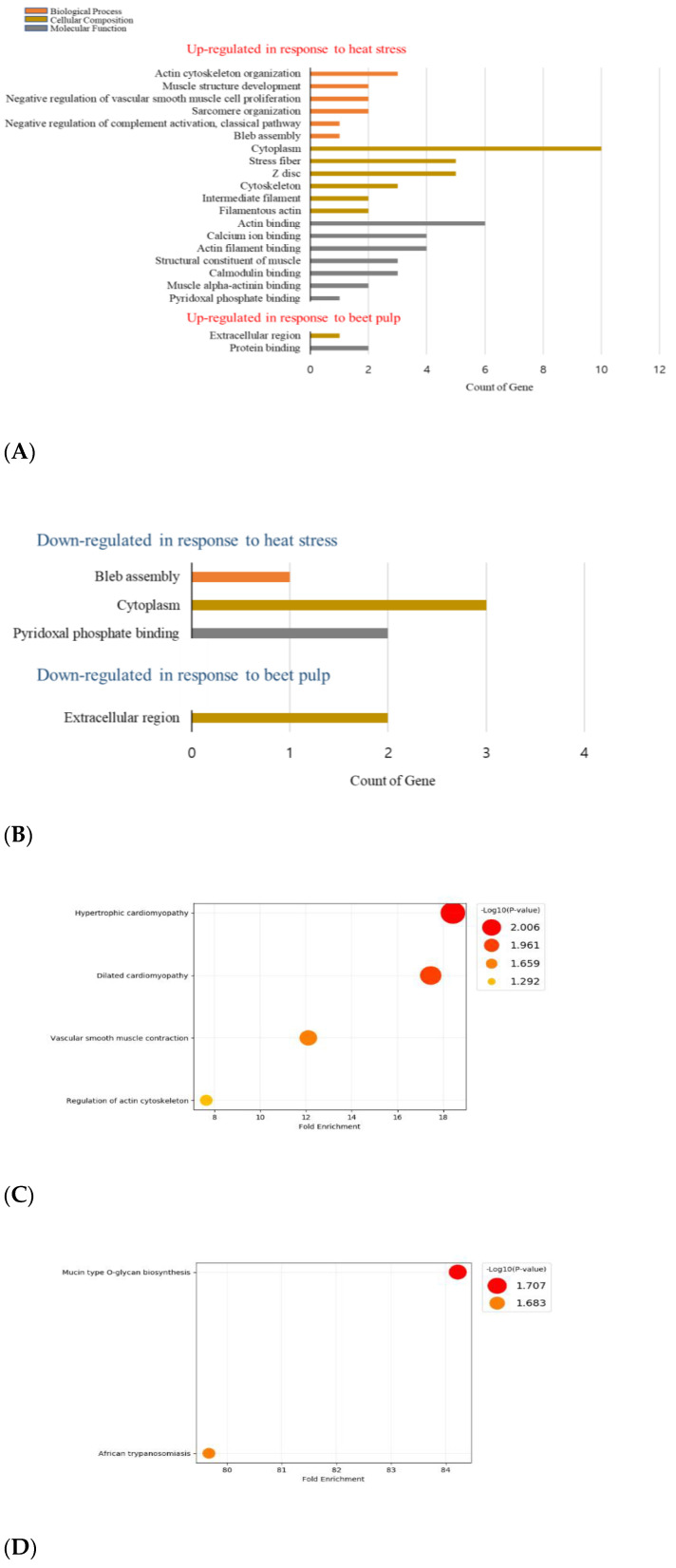
Functional analysis of the DEGs in HS and HS + BP treatments: (**A**) GO classification of the DEGs upregulated; (**B**) GO classification of the DEGs downregulated; (**C**) KEGG pathway enrichment results of the DEGs in HS vs. NT; (**D**) KEGG pathway enrichment results of the DEGs HS + BP vs. HS; The GO analysis was performed under Biological Process, Cellular Composition, and Molecular Functions categories; NT, normal temperature; HS, heat stress; HS + BP, dietary beet pulp under heat stress.

**Table 1 genes-13-01456-t001:** Expression levels of differentially expressed genes in HS and BP groups. Red and blue background color in the fold change values means the increased and decreased gene expression, respectively, with the intensity of color indicating the magnitude of change.

Gene Symbol	Fold Change			Gene Expression Value			Gene Description
	HS/NT	HS + BP/NT	HS + BP/HS	Normalized Data (log2)			
				NT	HS	HS + BP	
*GPT2*	0.472	0.452	0.957	4.658	3.575	3.511	Glutamic-pyruvic transaminase 2
*HSPB6*	2.191	2.629	1.200	6.224	7.356	7.618	Heat shock protein family B (small) member 6
*PPP1R14A*	2.194	2.337	1.065	5.389	6.522	6.613	Protein phosphatase 1 regulatory inhibitor subunit 14A
*CKM*	3.128	4.095	1.309	3.706	5.352	5.740	Creatine kinase, M-type
*SYNM*	2.001	2.743	1.371	5.551	6.552	7.007	Synemin
*NEXN*	2.026	2.077	1.025	3.100	4.119	4.155	Nexilin F-actin binding protein
*TSPAN1*	0.418	0.471	1.126	6.131	4.872	5.044	Tetraspanin 1
*GSTA1*	2.488	2.405	0.966	6.016	7.331	7.282	Glutathione S-transferase α 1
*GZMB*	0.374	0.432	1.155	7.639	6.219	6.427	Granzyme B
*TAGLN*	2.363	2.987	1.264	8.874	10.115	10.453	Transgelin
*C4BPA*	2.145	2.244	1.046	4.646	5.747	5.812	Complement component 4 binding protein, α
*DDC*	0.407	0.494	1.213	5.162	3.866	4.145	Dopa decarboxylase
*CSRP1*	2.047	2.352	1.149	6.886	7.919	8.120	Cysteine and glycine rich protein 1
*TPM2*	2.377	2.823	1.188	8.506	9.755	10.003	Tropomyosin 2 (β)
*CCL4*	0.467	0.442	0.945	4.123	3.026	2.945	C-C motif chemokine ligand 4
*ITGAE*	0.431	0.483	1.121	4.612	3.398	3.563	Integrin subunit α E
*PMP22*	0.406	0.436	1.073	6.643	5.343	5.445	Peripheral myelin protein 22
*MYLK*	2.216	2.703	1.220	6.927	8.074	8.361	Myosin light chain kinase
*SH3BGR*	2.097	2.377	1.133	3.716	4.784	4.965	SH3 domain binding glutamate rich protein
*PCP4*	2.787	3.488	1.251	5.749	7.228	7.552	Purkinje cell protein 4
*MX1*	0.356	0.408	1.147	6.084	4.593	4.791	MX dynamin like GTPase 1
*C14H10orf99*	0.452	0.417	0.924	5.340	4.194	4.079	Chromosome 14 C10orf99 homolog
*PDLIM3*	2.017	2.073	1.028	4.060	5.072	5.112	PDZ and LIM domain 3
*DES*	2.696	3.604	1.337	7.855	9.286	9.705	Desmin
*MYL9*	2.248	2.968	1.320	7.815	8.984	9.385	Myosin light chain 9
*PYGM*	2.173	2.416	1.112	3.896	5.016	5.169	Glycogen phosphorylase, muscle associated
*CNN1*	2.485	3.221	1.296	7.535	8.849	9.223	Calponin 1
*PDLIM7*	2.001	2.276	1.137	5.052	6.052	6.238	PDZ and LIM domain 7
*GNLY*	0.317	0.356	1.123	6.921	5.265	5.432	Granulysin
*ACTG2*	3.296	4.261	1.293	8.105	9.825	10.196	Actin γ 2, smooth muscle
*PLB1*	0.468	0.407	0.869	5.122	4.028	3.825	Phospholipase B1
*S100A2*	2.226	6.477	2.910	4.019	5.173	6.714	S100 calcium binding protein A2
*TPM1*	2.190	2.943	1.344	8.031	9.162	9.588	Tropomyosin 1 (α)
*MGP*	2.374	2.155	0.908	7.204	8.451	8.311	Matrix Gla protein
*CYP2J34*	0.837	1.703	2.034	3.882	3.626	4.650	Cytochrome P450 family 2 subfamily J member 34
*HBB*	1.693	3.886	2.295	7.759	8.519	9.717	Hemoglobin, β
*SCGB1A1*	1.291	0.327	0.253	3.898	4.266	2.286	Secretoglobin family 1A member 1
*LOC396781*	1.625	0.555	0.341	9.311	10.011	8.460	IgG heavy chain
*SAA3*	85.820	1.642	0.019	0.588	7.012	1.304	Serum amyloid A-3 protein
*ST3GAL1*	0.618	0.089	0.144	6.072	5.378	2.584	ST3 β-galactoside α-2,3-sialyltransferase 1
*HSPA6*	1.163	3.423	2.944	3.008	3.226	4.784	Heat shock protein family A (Hsp70) member 6
*S100A2*	2.226	6.477	2.910	4.019	5.173	6.714	S100 calcium binding protein A2
*GCNT3*	0.258	0.526	2.042	5.071	3.115	4.145	Glucosaminyl (N-acetyl) transferase 3, mucin type
*LYZ*	1.094	2.471	2.259	10.398	10.527	11.703	Lysozyme

## Data Availability

Not applicable.

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
