# Peer review of "Transcriptomic Analysis of the Porcine Gut in Response to Heat Stress and Dietary Soluble Fiber from Beet Pulp"

_genes, 2022, doi:10.3390/genes13081456_

Round 1

Reviewer 1 Report

The main aim of the manuscript “Transcriptomic analysis of the porcine gut in response to heat stress and dietary soluble fiber from beet pulp” was to study the impact of heat stress (HS) and the effects of dietary soluble fiber from beet pulp (BP) on gene expression of the porcine jejunum using RNAseq approach.

I recommended that the following comments and suggestions could taking into account before publish:

Keyword. A word related with fiber diet could be included.

Included the used reference genome in section “2.6. Bioinformatics Analysis”.

When the authors mentioned the log change is the log2 fold change. Please clarify.

In line 115.  Is p-values or adjusted p-values?

The thresholds of fold-changes mentioned in lines 115 and 116 are confusing.

The PCA was performed using the normalized counts or DGEs among samples?

The section “3.2. Porcine Gut Transcriptome Response to HS” could be reduced and mentioned only some examples of gene and go categories and include the full data in tables or figures. The same comment could be applied to section “3.3. Porcine Gut Transcriptome Response to BP under HS”.

The results obtained do not show a clear effect of supplementation. In this sense, the PCA does not clearly differentiate the BT+HS and HS groups, and the number of DEGs is reduced between them (ten genes). This is not clearly mentioned in the discussion and in the conclusion sections.

Reviewer 2 Report

The authors investigated the impact of heat stress (HS) and the effects of dietary soluble fiber from beet pulp (BP) on gene expression in the jejunum of growing pigs using transcriptomic analysis. The authors observed that dietary BP could improve intestinal epithelium integrity and immune response to pathogens under the heat stress. However, there are some issues concerning the experimental design and interpretation of results.

1.     I concern about the limited replicates for analysis. The authors have to provide the reason why only 3 pigs in each group were slaughtered to collect the jejunum sample.

2.     The interpretation concerning the comparison in genes differentially expression between HS+BP and NT was lacked.

3.     A combined table that contains the gene expression in each groups (log2 ± SD) should be used to replace Tables 2 and 3, which could be more visible to understand the impact of HS and effect of BP.

Round 2

Reviewer 1 Report

The authors take into account all comments and suggestion made in the previous review-report. For this reason, I recomend that the present version of the manuscript can be publish in the journal Animals.

Reviewer 2 Report

The authors made a big efforts to revise the manuscript.